# Emerging Concepts of Targeted Protein Degrader Technologies via Lysosomal Pathways

**DOI:** 10.3390/ijms26125582

**Published:** 2025-06-11

**Authors:** Mohammad Maqusood Alam, Sobia Wasim, Sang-Yoon Lee

**Affiliations:** 1Neuroscience Research Institute, Gachon University, Incheon 20565, Republic of Korea; maqusood.in@gmail.com (M.M.A.); sobiawasim28@gmail.com (S.W.); 2Department of Neuroscience, College of Medicine, Gachon University, Incheon 21936, Republic of Korea

**Keywords:** target protein degrader (TPD), autophagy receptor, lysosomal degradation, extracellular protein, membrane protein, bifunctional antibody

## Abstract

Targeted protein degradation (TPD) has emerged as a revolutionary strategy for modulating protein function, offering a promising alternative to traditional small-molecule inhibitors. The distinctive mechanism of action in TPD has previously allowed researchers to target undruggable proteins, broadening the scope of “druggable” properties and expanding the scope of therapeutic possibilities. As the field of TPD advances, several alternative strategies to proteolysis-targeting chimeras (PROTACs) have emerged, which do not rely on the E3 ubiquitin ligase recruitment mechanism, expending the scope of TPD. Recently, several new technologies have emerged for TPD of extracellular and membrane proteins. While encouraging progress has been made in this field, the application of these technologies remains in its early stages. In this review, we explore the therapeutic potential of current key emerging lysosome-mediated TPD approaches by summarizing key discoveries and address the challenges associated with degrading extracellular and membrane protein targets. We also outline the chemical structure, activity, and pharmaceutical properties of each degrader, as well as the development of chemical probes for perturbing autophagy pathways.

## 1. Introduction

Targeted protein degradation (TPD) is an emerging therapeutic strategy in drug discovery, particularly for tackling disease-causing proteins that accumulate abnormally and are challenging to treat using existing inhibitors. These aberrant proteins are involved in various diseases, including neurodegenerative diseases and type 2 diabetes [1]. Compared to conventional small-molecule inhibitors and gene regulation technologies, TPD offers distinct advantages in selectivity, efficacy, and mechanism of action. Unlike traditional inhibitors, which require high-affinity binding and can lead to off-target effects and drug resistance, TPD enables the selective elimination of disease-causing proteins that were previously considered “undruggable”. TPD utilizes the cell’s own degradation systems to eliminate specific proteins, significantly expanding the range of druggable targets [2]. TPD primarily occurs through two key cellular pathways: the ubiquitin–proteasome system (UPS) and lysosomal pathways [3,4]. The UPS pathway responsible for the degradation of intracellular, soluble, and short-lived proteins tagged with ubiquitin [4,5], while lysosomal pathways are broadly categorized into two main types: the autophagy–lysosomal pathway and the endosome–lysosome pathway. Although both pathways deliver cargo to lysosomes for degradation, they differ in the origin of the cargo and the mechanism by which it is transported [6]. The autophagy–lysosomal pathway is a major intracellular degradation system that recruits targeted proteins to autophagosomes, which subsequently fuse with lysosomes for degradation, while endosomes–lysosomes are involved in the transport of extracellular material into the cell. Both pathways ultimately converge on lysosomes, where degradation occurs. The UPS, autophagy, and endosome–lysosomal pathways represent a promising platform for the precise and efficient targeting of diseases by enabling TPD [7,8]. The activity of these pathways is tightly regulated within the cell to maintain controlled protein degradation. PROTACs, introduced in 2001, represent a novel chemical biology strategy that utilizes the UPS for targeted protein degradation, which influences drug discovery [9,10]. These molecules are bifunctional, designed to bind both a target protein of interest (POI) at one end and recruit an E3 ubiquitin ligase on the other, connected by a diverse range of linkers [11,12]. PROTACs carry the E3 ligase to the transient ternary complex, triggering the UPS to degrade POI and its subsequent proteasomal degradation [13]. Several PROTAC molecules are currently under evaluation in clinical trials [14]. However, PROTAC approaches relying on the UPS pathway predominantly target intracellular, soluble, and short-lived proteins. These limitations hinder their potential application in diseases involving proteasome-resistant proteins, extracellular proteins, and aggregate proteins. Beyond UPS-based strategies, novel approaches are emerging that utilize the lysosomal pathway, offering a potential solution for targeting unparalleled proteins and organelles that are independent of proteasomes [15]. As an alternative protein degradation pathway, lysosomes serve as a viable mechanism for target proteins and organelles that are inaccessible to proteasomal degradation. Autophagy–lysosomal TPD, such as that involving autophagy-targeting chimeras (AUTACs), induces the degradation of target proteins through the selective autophagy pathway [16], and AUTOphagy-TArgeting Chimeras (AUTOTACs) bind directly to the ZZ domain of p62/SQSTM1 for target degradation [17]. AuTophagy-TEthering Compounds (ATTECs) represent an innovative class of biofunctional molecules that activate the autophagosomal pathway to degrade a variety of intracellular proteins and non-protein cellular components [18,19]. In addition, innovative technologies that utilize the endosome–lysosome pathway for targeted cellular degradation are emerging, including lysosome-targeting chimeras (LYTACs) [20,21] and MoDE-As (molecular degraders of extracellular proteins via the asialoglycoprotein receptor ASGPR), which efficiently target membrane-associated or extracellular proteins [22]. Transcytosis-inducing molecular degraders of extracellular proteins (TransMoDEs) are designed to degrade targeted proteins via lysosomes [23]. The integrin-facilitated lysosomal degradation (IFLD) strategy eliminates both extracellular and cell membrane proteins [24]. In addition, Antibody-based PROTAC (AbTAC) [25] and proteolysis-targeting antibodies (PROTAB) [26] facilitate transmembrane E3 ligase (RNF43 or ZNRF3)-mediated lysosomal degradation. GlueBody Chimeras (GlueTACs) utilize Cell-Penetrating Peptide-Lysosome-Sorting Sequence (CPP-LSS)-mediated internalization and lysosomal degradation [27]. Cytokine receptor-targeting chimeras (KineTACs) induce lysosomal degradation via cytokine-mediated internalization [28], Transferrin receptor-targeting chimeras (TransTACs) target membrane protein degradation [29], autophagy receptor-inspired targeting chimeras (AceTACs) degraders successfully targeted degradation of aggregation-pron proteins and protein aggregates [30], and Folate Receptor-Targeting Chimeras (FRTACs) selectively degrade both soluble and membrane-bound cancer-related proteins efficiently [31]. These novel therapeutic approaches have potential for small-molecule drug discovery beyond conventional inhibition or antagonism. However, these innovative technologies still require significant development to establish their capabilities as platform technologies. This review explores key emerging lysosome-mediated TPD technologies for the development of degradation strategies toward extracellular and membrane proteins, including their applications and the therapeutic potential of recently developed chemical probes, advantages, and their limitations.

## 2. An Overview of the History of TPD

TPD is mediated by the UPS and the autophagy–lysosomal pathways, where proteasomal and lysosomal enzymes degrade targeted proteins through autophagy. Together, these pathways form a novel platform for precisely and effectively targeting disease-associated proteins, including previously “undruggable” targets. Since the first PROTAC was reported by Sakamoto et al. in 2001 [12], subsequent advancements in drug design have enhanced efficiency against target proteins. Emerging lysosome-dependent TPD technologies, such as AUTOTAC and ATTEC, were reported in 2019, and LYTAC was reported in 2020 (Figure 1). Most TPD strategies, such as PROTACs and molecular glues, utilize the UPS pathway to degrade target proteins. In contrast, lysosome-dependent TPD strategies, such as chaperone-mediated autophagy (CMA)-based degraders, AUTACs, AUTOTACs, and ATTECs, rely on the autophagy–lysosomal pathway to target membrane proteins, intracellular proteins, and protein aggregates. Other approaches, including LYTACs [20,21], MoDE-As [22], AbTACs [25], PROTABs [26], bispecific aptamers [32], GlueTACs [27], KineTACs [28], TransTACs [29], AceTACs [30], and FRTACs [31], engage the endosome/autophagy–lysosomal pathway to address membrane proteins, extracellular proteins, and protein aggregates, significantly broadening the spectrum of druggable substrates.

## 3. New Frontiers in Lysosome-Mediated TPD

While PROTAC technology is very promising, its alliance with specific E3 ligases and the proteasomal pathway may restrict its effectiveness in certain cell types or against proteasome-resistant proteins. In recent years, the lysosomal degradation pathway—an essential degradation system independent of proteasome—has emerged as a key focus in TPD strategies [7,33,34]. This pathway includes both autophagy and the endosome/lysosome pathways [35,36]. Autophagy, a key cellular process, maintains homeostasis and supports cell survival by degrading and recycling damaged organelles and dysfunctional proteins. Autophagy dysregulation has been linked to various diseases, including cancer and neurodegenerative disorders. In mammalian cells, autophagy occurs in three forms, namely macroautophagy, microautophagy, and CMA, each utilizing unique mechanisms for lysosomal degradation. The autophagy process begins with a membrane structure known as a phagophore, which is formed from lipid bilayers with lipidated LC3 proteins. Phagosomes expand to sequester intracellular components, creating an autophagosome that subsequently fuses with the lysosome for degradation. In the endosome/lysosome pathway, material is processed through a series of membrane-bound compartments. These compartments involving endosomes and lysosomes, facilitate the early endosome carrier vesicles, late endosomes, and ultimately fuse with the lysosome, where materials are degraded. Both pathways effectively degrade cellular components, and innovative strategies are emerging to selectively harness each pathway for therapeutic purposes.

## 4. Emerging Lysosome-Based Targeted Protein Degradation

TPD technologies have been extensively explored through PROTACs as therapeutics or chemical probes. However, these technologies still face certain limitations. Some emerging technologies based on lysosome mechanisms, such as AUTACs, AUTOTACs, ATTECs, LYTACs, MoDE-As, TransMoDEs, IFLDs, GlueTACs, KineTACs, TransTACs, AceTACs, and FRTACs, are under development. These novel chemical biology platforms can address the limitations of PROTAC technology. In this work, we explore the current advancements in lysosome-based degraders for targeting various proteins (Table 1).

### 4.1. AUTACs

Arimoto’s group developed the AUTAC system, a novel degradation technology that utilizes autophagy for protein degradation [16,37,42,43]. Although AUTACs share conceptual similarities with PROTAC technology and operate through ubiquitination, they have key differences in mechanisms. For example, PROTACs link POI to an E3 ligase subunit to initiate K48 polyubiquitination, whereas AUTACs promote K63 polyubiquitination, which is recognized by the selective autophagy pathway, leading to the degradation of the target POI [16]. AUTAC tehnology utilizes modified guanine as a degradation tag, a warhead to target POI, and a flexible linker to bind specific target (Figure 2A) [16,42]. Studies have demonstrated that when S-guanylation was induced by a synthetic guanine derivative, p-fluorobenzyl guanine (FBnG), this effectively led to autophagic degradation. To validate this hypothesis, Arimoto’s team developed the AUTAC platform, which degrades endogenous proteins by linking a degradation tag (like FBnG) to a ligand specific to the POI, promoting selective autophagy through induced K63-polyubiquitination [16,42]. Using this approach, they designed AUTAC degraders targeting MetAP2, FKBP12, BRD4, and mitochondria. Notably, mitochondria-targeting AUTACs (mito-AUTACs) facilitated the removal of damage and fragmented mitochondria through mitophagy [37,43]. AUTAC-1 provided a proof of concept by targeting the endogenous protein methionine aminopeptidase 2 (MetAP2) by using an S-guanylation tag for autophagic degradation. AUTAC1 utilizes Fumagillol as a warhead, which covalently binds and degrades MetAP2, significantly reducing its levels at concentrations of 1–100 µM. The experimental findings confirm that AUTAC-1 successfully reduced the MetAP2 protein through the autophagy–lysosomal degradation pathway [16]. AUTAC-2 is a non-covalent ligand designed to target FKBP12, a receptor for the immunosuppressant FK506-binding protein. The structure of AUTAC2 contains an FBnG group and an SLF moiety. HeLa cells were treated with 10–100 µM of AUTAC-2, and FKBP12 was effectively degraded, demonstrating the potential of non-covalent ligands in AUTAC technology [16]. However, the dependence on autophagy was not evaluated in this study.

In a separate design, AUTAC-3 was developed using the epigenetic anticancer agent JQ-1 acid as the warhead to target the BET protein family, particularly Brd4. Brd4 is essential for melanoma survival but challenging to target due to its nuclear localization. Treatment with increasing concentrations of AUTAC-3 effectively reduced Brd4 levels, highlighting the potential of AUTACs to target intracellular proteins [16]. Takahashi et al. demonstrated AUTAC4 for selective degradation of dysfunctional mitochondria through mitophagy [16,42]. The guanine moiety in AUTAC4 interacts with mitochondrial translocator protein (TSPO) on the outer mitochondrial membrane, which promotes the selective degradation of dysfunctional mitochondria. In earlier work, the same group introduced AUTAC2, the first generation of autophagy-based degraders. To enhance its efficacy, they explored alternative substructures to cysteine through structure–activity relationship (SAR) studies. These investigations revealed that replacing cysteine with alternative moieties significantly improved degradation activity. Notably, they identified 2G-AUTAC2, a second-generation AUTAC with a 100-fold increase in activity compared to the first-generation AUTAC2, demonstrating effectiveness at a 10 µM concentration [37]. This demonstrates the potential of AUTACs as a powerful tool for TPD in drug development. The chemical structures of designed AUTACs 1, 2, 3, and 4 and 2G-AUTAC2 are illustrated in Figure 2B. The AUTAC system utilizes the S-guanylation tag to harness a ubiquitination-dependent pathway for lysosomal degradation. However, a significant limitation of AUTAC technology is that the mechanism by which S-guanylation induces K63 polyubiquitination remains unclear.

### 4.2. AUTOTACs

AUTOTAC is a platform technology that utilizes bifunctional molecules that combine target-binding ligands (TBLs) with autophagy-targeting ligands (ATLs) [17,44,45]. Once the TBL binds to its target, ATLs interact with the ZZ domain of the p62/SQSTM1 (sequestosome 1), triggering p62 oligomerization and the formation of the target-p62 complex (Figure 3A) [17]. This process reveals the LC3-interacting region (LIR) domain of P62, facilitating oligomerization and subsequent autophagic targeting and lysosomal degradation through a multi-step mechanism. AUTOTAC has demonstrated efficacy in degrading various oncoproteins and resistant aggregates in both in vitro and in vivo models, achieving DC_50_ values in the nanomolar range [44]. To establish AUTOTAC as a versatile degradation platform, Ji et al. developed AUTOTACs to target soluble protein such as estrogen receptor beta (ERβ), androgen receptor (AR), and methionine aminopeptidase 2 (MetAP2) [17]; these cancer targets were previously explored in PROTAC research [46].

These AUTOTACs were designed by conjugating the known ligands of these targets to discovered ATLs. For instance, an AUTOTAC based on the nonsteroidal synthetic ERβ ligand PHTPP-1304 induced the degradation of ERβ in HEK 293T cells and MCF-7 cells with DC_50_ values of 2 nM and under 100 nM, respectively [17]. Notably, neither ATLs nor TBLs alone induce degradation, indicating that ATL-induced p62/SQSTM1 activation alone is inadequate for target degradation. Similarly, AR-targeting AUTOTACs, such as vinclozolin M2-2204, triggered AR degradation with a DC_50_ value of around 200 nM in LNCaP cells, although Fumagillin-105 targeted MetAP2 degradation with a DC_50_ value of about 500 nM in U87-MG cells [46].

To address the challenge of UPS-resistant misfolded and aggregated proteins, which contribute to neurodegenerative diseases like Alzheimer’s disease pathology, such as aggregation-prone P301L tau mutants, misfolded proteins and aggregates were targeted using two specific ligands: 4-phenylbutyric acid (PBA) and Anle138b. PBA binds to the exposed hydrophobic region of misfolded proteins, while Anle138b specifically targets oligomeric misfolded proteins. These ligands were conjugated with p62-binding moieties to develop four AUTOTAC compounds: PBA-1105, PBA-1106, Anle138b-F105 (ATC 161), and PBA-1105b. PBA-1105 and PBA-1106 are designed to target mutant tau proteins, PBA-1105b targets mutant desmin proteins, and Anle138b-F105 focuses on mutant huntingtin protein, facilitating their clearance through macroautophagy [17]. Notably, all of these AUTOTACs induced the degradation of mutant desminL385P but not wild-type desmin. PBA-1105 and PBA-1106 can induce autophagic degradation of the aggregated expressed mutant tau protein with a DC_50_ value in the range of 1–10 nM. The effectiveness of AUTOTAC-mediated degradation seems to be largely independent of linker length. This was demonstrated by PBA-1105b, which utilizes a much longer PEG-based linker than PBA-1105 while maintaining strong anti-tauP301L degradative activity. Additionally, Anle138b-F105 (ATC161) selectively induced the degradation of α-syn aggregates at a DC_50_ value of approximately 100 nM [46]. They also degraded the aggregation-prone P301L mutant tau protein with DC_50_ values as low as 3 nM [17]. The chemical structures of several designed AUTOTACs are illustrated in Figure 3B,C. These findings confirm the strong capacity of AUTOTACs to selectively degrade both intracellular proteins and protein aggregates.

### 4.3. ATTECs

ATTEC technology has emerged as a potent approach for TPD [18,19]. Unlike PROTACs and AUTACs, ATTECs operate independently of ubiquitination. These bifunctional molecules bridge the POI to autophagosomes by directly binding both the POI and the essential autophagosome protein LC3 (Figure 4A). A proof-of-concept study utilized small-molecule microarray-based screening to discover compounds that interact with LC3 and mutant huntingtin (mHTT) protein, which contains an expanded polyglutamine (polyQ) sequence linked to Huntington’s disease [47]. These compounds specifically bind to mHTT and not to wild-type HTT, and they reduce other polyQ-expanded proteins such as mutant ataxin-3 (ATXN3). Once the ATTEC molecule binds to the autophagosome via LC3, these polyQ proteins are tagged for degradation.

Recently, Lu’s group extended the ATTEC platform to target non-protein molecules, particularly lipid droplets (LDs), resulting in the creation of LD-ATTEC1, LD-ATTEC2, LD-ATTEC3, and LD-ATTEC4 [19]. LDs, which store and release lipids within cells, can undergo autophagic degradation. Unlike PROTACs and AUTACs, which are unable to target lipids, LD-ATTECs link lipid-sensing probes to LC3-binding components, facilitating the precise breakdown of lipid droplets without affecting other lipid-rich membranes and overall autophagy processes [19]. This significant advancement in ATTEC technology expands its capabilities beyond protein degradation to target non-protein biomolecules like lipids. This innovative strategy offers potential therapeutic applications for treating metabolic conditions, such as obesity, fatty liver disease, and neurodegeneration, which are characterized by excessive LD accumulation. This strategy could be applied to various types of target cells.

The identification of potent LC3 binders has prompted further exploration of autophagy for targeted intracellular protein degradation. Several ATTEC molecules were effectively targeted to degrade oncoprotein, including bromodomain-containing protein 4 (BRD4) [38] and nicotinamide phosphoribosyl transferase (NAMPT) [39]. For instance, Pei et al. developed an LC3-based autophagy chimeric ATTEC molecule named compound 10f that successfully degraded the BRD4 protein through the autophagic pathway and demonstrated potent anti-proliferative activity in several tumor cells. Compound 10f achieved high degradation efficiency at a concentration of 0.5 µM in MDA-MB-231 and MDA-MB-468 cells [38]. Similarly, Dong et al. [39] developed the ATTEC approach to create the first autophagic degraders of NAMPT. By linking a NAMPT inhibitor to an LC3-binding element (Ispinesib) using a flexible linker, they generated a NAMPT-ATTEC molecule named compound A3. These results demonstrate that compound A3 efficiently degrades NAMPT via the autophagy–lysosome pathway, offering a new strategy for the targeted degradation of NAMPT [39].

Recently, Tan et al. [40] utilized the previously mentioned LC3 ligand GW to link with the translocator protein (TSPO) ligand, creating an ATTEC molecule named mT1, which is capable of targeting damaged mitochondria for clearance [40]. Compound mT1 was found to promote mitochondrial degradation through LC3 and autophagy-related proteins (ATGs), mitigating disease phenotypes in Parkinson’s disease (PD) cell models and Down syndrome (DS) organ models [40]. The chemical structures of the designed ATTECs (C1~C4) and compounds 10f, A3, and mT1 are illustrated in Figure 4B. Studies have demonstrated the potential of ATTEC molecules in targeting mitochondria, including their ability to degrade organelles, offering a new strategy for investigating mitochondria-related diseases. However, further studies are needed to refine these mechanisms and explore the potential of other chimeric molecules in targeted degradation strategies.

### 4.4. LYTACs

Extracellular proteins, such as growth factors, cytokines, immune complexes, and others, have a critical role in disease progression. However, their extracellular location makes them inaccessible to the UPS, making them unsuitable for being targeted by PROTACs [48]. To address this limitation, LYTACs were developed, enabling the degradation of extracellular and membrane-bound proteins, expanding the scope of potential targeted protein degradation [49]. The Bertozzi group developed LYTACs to degrade extracellular secreted proteins and membrane-associated proteins [20,21]. LYTACs are bifunctional molecules comprising a target binding moiety, such as a small molecule or antibody, linked to a glycan ligand that binds to lysosome-targeting receptors, including cation-independent CI-M6PR or the asialoglycoprotein receptor (ASGPR) for liver-specific degradation (Figure 5A) [21]. The mannose 6-phosphate (M6P) receptor such as Cl-M6PR were discovered their role in transporting M6P-contaning soluble acid hydrolyses from the Golgi to lysosomes. This receptor can also regulate cell growth, motility, and may act as a tumor suppresser [50]. First-generation LYTACs utilized CI-M6PR to deliver target protein to the lysosome for degradation [20]. These LYTACs employed a target-specific antibody or inhibitor molecule conjugated to a mannose-6-phosphonate oligopeptide that acts as an agonist for CI-M6PR. The interaction directed the POI, including apolipoprotein E4, EGFR, CD71, and PD-L1 to CI-M6PR, promoting its internalization and the degradation of the lysosome, while CI-M6PR was recycled back to the membrane [20]. To monitor uptake, neutravidin (NA), a fluorescently labeled protein stable in endosomal and lysosomal conditions, was paired with the LYTAC molecule. SAR studies have shown that ligand valencies, conjugation sites, and the different modalities of POI binders can influence LYTAC’s effectiveness.

Second-generation LYTACs harnessed the liver-specific ASGPR, which is exclusively expressed in hepatocytes, allowing for a tissue-specific strategy for protein degradation. These LYTACs were connected to ligands such as Galactosamine (Gal) or N-acetylgalactosamine (GalNAc). Trivalent ligands (tri-GalNAc) demonstrated superior receptor binding and therefore showed superior efficacy. They linked the ASGPR ligand (tri-GalNAc) to biotin and an antibody to generate a diverse class of degrader (tri-GalNAc-biotin) that could internalize and degrade neutrAvidin and EGFR in liver cells [41]. Additionally, the optimization of tri-GalNAc-LYTACs by conjugating them to antibody scaffolds, which attenuated EGFR signaling, reduced cancer cell proliferation and exhibited improved pharmacokinetic properties. This platform highlights a promising approach for liver-targeted and cell-type-specific protein degradation [21]. In parallel, Wu et al. [51] further explored ASGPRs by developing an aptamer-based LYTAC (Apt-LYTAC) system. This approach involved conjugating a target-specific RNA aptamer to tri-GalNAc, enabling the liver-cell-specific degradation of disease-relevant extracellular and membrane protein. The Apt-LYTAC demonstrated nanomolar affinity for ASGPRs, facilitating clathrin-mediated endocytosis of the receptor–ligand. This Apt-LYTAC platform efficiently degrades proteins like PDGF and PTK7 via the lysosomal degradation pathway, which plays a crucial role in cancer metastasis and proliferation.

### 4.5. MoDE-As, TransMoDEs, and IFLDs

Following a similar approach, the Spiegel group developed a modular, bifunctional synthetic molecule called MoDE-As, which is specifically design to target ASGPRs and induce the degradation of extracellular proteins [22]. MoDE-As molecules successfully targeted and degraded protein like the α-DNP antibody and the cytokine MIF. These molecules rely on ASGPR internalization through the clathrin-mediated endocytosis and lysosomal pathway, offering a modular strategy for targeting extracellular proteins. Importantly, the treatment of LYTACs did not impair lysosomal function, suggesting minimal cellular toxicity. However, a significant drawback is the large size of LYTAC molecules, which may prevent them from penetrating intracellular targets, limiting their applicability. In contrast, MoDE-As, as small molecules, may offer better tissue penetration compared to antibody-based LYTACs. The chemical structures of designed LYTACs and MoDE-As are illustrated in Figure 5B.

**TransMoDEs:** TPD emerged as promising therapeutic approach for various diseases; however, its application in the central nervous system (CNS) is restricted by the limitation of the blood–brain barrier (BBB). Howell et al. [23] presented a new class of bifunctional small molecules called TransMoDEs designed to both degrade target proteins via lysosomes and facilitate their transport protein across the brain’s endothelial cells. These degraders are derived from Angiopep-2, a peptide known for its ability to cross the BBB through receptor-mediated transcytosis. Studies have demonstrated that TransMoDEs conjugated with either biotin or chloroalkane ligands can induce the uptake of streptavidin or HaloTag proteins, respectively. Interestingly, while lipoprotein receptor-related protein 1 (LRP1) is known as the main receptor for Angiopep-2, TransMoDE-mediated uptake does not solely depend on this pathway [23]. Further investigation using a BBB model revealed that TransMoDE-driven endocytosis of streptavidin follows a clathrin-mediated mechanism, leading to both lysosomal degradation and the transcytosis of the target protein. These findings demonstrate that TransMoDEs can effectively recruit target proteins, transport across the BBB, and degrade proteins of interest in CNS-relevant cells. These findings support the continued development of TransMoDEs as a potential strategy for eliminating harmful proteins in neurodegenerative diseases [23].

**IFLDs:** Zheng et al. [24] introduced a new approach called the IFLD strategy for eliminating both extracellular and cell membrane proteins using bifunctional molecules as degraders. These molecular degraders can induce the endocytosis and lysosomal degradation of target proteins through the formation of a three-part complex involving the protein of interest and integrins on the cell surface. The αvβ3 integrin is often overexpressed in tumors and is a target for the IFLD strategy. This approach is particularly well suited for cancer-specific protein degradation. For instance, the BMS-L1-RGD compound effectively degraded PD-L1, presenting a promising approach for cancer-specific protein degradation [24]. Compared to technologies relying on antibodies, nanobodies, and aptamers, bifunctional small molecules offer unique benefits as IFLD protein degraders, including a lack of immunogenicity and the ability to control their pharmacological and pharmacokinetic properties due to their small size [24]. Moreover, the existence of small molecule inhibitors for many disease-related proteins simplifies the design of IFLD molecular degraders.

In summary, well-established PROTAC technology and its further advancement, including at least five emerging autophagy–lysosome TPD technologies, was demonstrated (Figure 2A, Figure 3A, Figure 4A and Figure 5A). Although each approach presents unique advantages and limitations (Table 2), they significantly broaden the scope of degradation applications and may open a new avenue to revolutionize the field of targeted protein degradation.

### 4.6. AbTACs and PROTABs

AbTACs or PROTABs are innovative degradation platforms that employ bispecific antibodies to bind a plasma membrane E3 ligase with one arm, whereas the other arm binds to other targets a POI, thereby bringing the two into close proximity to enable ubiquitin-mediated degradation. AbTACs have been demonstrated to successfully target membrane proteins such as PD-L1 and EGFR for degradation by utilizing specific E3 ligases. PD-L1 is degraded via E3 ligase RNF43 (ring finger protein 43) [26], while EGFR is targeted through ZNRF3 (zinc and ring finger 3) [52]. These E3 ligases are transmembrane proteins that play a pivotal role in ubiquitination, which is the most important pathway that regulates protein degradation in cells. In parallel, PROTABs have induced the degradation of membrane proteins, including IGF1R, HER2, and PD-L1, by hijacking up to five different transmembrane E3 ligases [53]. Notably, the E3 ligases RNF43 and ZNRF3 act as negative regulators of the Wnt signaling pathway and are upregulated in colorectal cancer (CRC) [54]. An SAR analysis of AbTACs and PROTAB molecules suggests that their efficacy depends on the geometry orientation and stability of the ternary complex, much like small-molecule PROTACs.

### 4.7. GlueTACs

GlueTACs represent an emerging class of protein degraders that operate independently of endogenous cellular effectors, distinguishing them from LYTACs, KineTACs, and TransTACs. GlueTAC is a covalent nanobody-based PROTAC strategy designed for efficient targeted membrane protein degradation and high specificity. The irreversible binding of covalent nanobodies to surface antigens avoids off-target effects during endocytosis and degradation. This process can be further accelerated by conjugation with a cell-penetrating peptide and a lysosomal-sorting sequence, causing the GlueTAC molecule to successfully trigger the internalization and degradation of PD-L1, offering a new avenue to target and eliminate cell-surface proteins [27]. This feature suggests potential broad applicability across diverse cell types regardless of biological effector expression patterns.

### 4.8. KineTACs

KineTACs represent a novel class of bifunctional antibody-based degraders that exploit cytokine receptor-mediated internalization to facilitate targeted protein degradation. These bifunctional chimeric molecules consist of a cytokine moiety that binds to its cytokine receptor and an antibody arm that recognizes the POI [28]. When the endogenous cytokine arm of the KineTAC binds to cytokine receptor, the entire KineTAC-bound POI complex undergoes endocytosis, directing the POI to the lysosome for degradation. To selectively degrade extracellular and membrane proteins, KineTACs provide a flexible and modular platform [29]. First-generation KineTACs utilized the C-X-C motif chemokine ligand 12 (CXCL12), which targets chemokine receptor type 7 (CXCR7) and receptor type 4 (CXCR4), and they successfully degraded a wide range of membrane proteins such as programmed cell death protein-1 (PD-1), PD-L1, EGFR, human epidermal growth factor receptor 2 (HER2), CUB domain-containing protein 1 (CDCP1), and trophoblast cell surface antigen 2 (TROP2). Unlike ubiquitin-dependent degradation approaches, KineTACs can also target soluble extracellular proteins like VEGF and TNF-α. The KineTAC structure relates to its activity (SAR), indicating that degradation efficiency depends more on the antibody’s affinity of binding to the POI than that to the cytokine receptor, with slower dissociation rates leading to better degradation [28]. The expansion of the KineTAC platform to additional cytokine receptors, such as IL-2R, has demonstrated its versatility. KineTACs represent a simpler and more flexible alternative to LYTACs, with significant promise for therapeutic applications.

### 4.9. TransTACs

TransTACs are bispecific molecules designed to cross the BBB through the transcytosis-mediated transferrin receptor (TfR). Their dual binding to both the TfR1 and POI facilitates the internalization of the complex, directing the target protein to lysosomes for degradation [29]. Cancer cells overexpress TfR1 due to their high iron for rapid growth and spread, making the TransTAC platform preferentially target cancer cells. TransTACs can effectively degrade various membrane proteins, including EGFR, CD20, PD-L1, and CD19-targeted CAR. They have demonstrated the capacity to control CAR-T cells by degrading CAR proteins, which may mitigate adverse effects such as cytokine release syndrome [55]. The cleavable linker and synthetic anti-TfR1 binder are vital components of TransTAC design. An SAR analysis revealed that TransTAC degradation efficiency is influenced by both the geometry of the TransTAC and the format of the POI binders. Moreover, studies have shown that TransTACs have demonstrated reversible control of human primary chimeric antigen receptor T cells and the targeting of drug-resistant EGFR-driven lung cancer in mouse xenograft models [56]. By exploiting the natural recycling pathway of TfR1, TransTACs facilitate the effective uptake and degradation of target proteins. This novel approach offers a powerful strategy for targeting membrane proteins that are difficult to address with conventional small-molecule inhibitors, offering a new avenue for innovative therapeutic interventions.

### 4.10. AceTACs

Jiang et al. [30] created autophagy receptor-inspired targeting chimeras (AceTACs) by conjugating autophagy receptor LIR domains to antibodies. These antibody-fusion-based degraders, which mimic autophagy receptors by simultaneously binding both LC3 and specific cellular targets on autophagosomes, facilitate lysosomal degradation. AceTACs efficiently degrade aggregation-prone proteins such as mutant huntingtin (mHTT), TAR DNA-binding protein 43 (TDP-43), and FUS mutants and organelles such as mitochondria, peroxisomes, and the ER [30]. Optimized AceTACs, especially with the LC3-interacting region of TP53INP2 and ALFA-targeting nanobodies, enhanced intracellular protein degradation efficiency, achieving up to 90% reduction in mHTT levels [57]. By decorating organelle membranes with recognition sites, AceTACs can induce targeted organelle aggregation and degradation. Recently, Hu et al. discovered a potent new compound, p53Y220C-specific AceTAC (MS182), which links to selective small-molecule binders of p53Y220C to acetylate p53Y220C in endogenous cellular systems [58]. MS78 suppresses the proliferation and clonogenicity of p53Y220C-mutant cancer cells. Furthermore, it shows minimal toxicity in cells with wild-type p53 and normal cells. MS182 represents a promising chemical tool to investigate the role of p53Y220C acetylation in cancer. However, despite their potential, current versions of AceTACs lack the ability to distinguish between healthy and damaged organelles, posing a challenge for therapeutic applications. Compared to other autophagy-based strategies like AUTOTACs, AceTACs offer a systematic approach to organelle degradation in mammalian cells, with specificity that can be further refined using techniques like phage display. Future research may expand AceTACs’ utility in viral degradation, such as SARS-CoV-2, by leveraging their role in autophagy activation [57]. Overall, AceTAC degraders show a promising strategy for treating neurodegenerative diseases and other cellular pathologies through selective degradation.

### 4.11. FRTACs

Zhou et al. [31] developed a new type of molecule called Folate Receptor-Targeting Chimeras (FRTACs). These FRTACs exploit the folate receptor, predominantly expressed on the surface of cancer cells, to selectively degrade both soluble and membrane-bound cancer-related proteins. Studies have demonstrated that using folate receptor- mediated TPD to selectively degrade proteins in many types of tumors makes them a promising target for cancer drug delivery. This FR-mediated TPD platform offers a potential alternative to existing therapies, such as immune checkpoint blockade (ICB), with the promise of improved precision and efficiency [31]. Overall, this work expands the possibilities of TPD by introducing tumor-selective targeting, and FRTACs could become a versatile, effective, and easily accessible technology for degrading oncogenic protein targets.

## 5. Conclusions and Future Perspectives

The precise regulation of protein degradation is essential for cellular function, as misfolded or unwanted proteins must be efficiently cleared to prevent cellular dysfunction. TPD has emerged as a promising therapeutic strategy for removing disease-related proteins. PROTACs have been extensively studied and operate by primarily relying on the UPS pathway for protein degradation. However, they are restricted to degrading soluble and membrane-bound proteins. Compared to PROTACs, which often exhibit nanomolar potency, lysosome-mediated TPD agents generally exhibit lower potency, likely due to differences in degradation kinetics between the UPS and the lysosome degradation pathway. Nevertheless, lysosome-based degradation agents may offer unique advantages by targeting misfolded proteins, aggregates, and dysfunctional organelles that are not amenable to UPS-mediated degradation. Lysosome-mediated TPD technology, such as AUTAC, AUTOTAC, ATTEC, LYTAC, MoDE-A, AbTACs, PROTAB, bispecific aptamer, KineTAC, TransTAC, AceTAC, and FRTAC, overcomes some of the limitations of proteasome-mediated degradation by leveraging the autophagy and endosome–lysosome pathways to selectively degrade pathogenic proteins and organelles, thereby expanding the range of therapeutic targets. For example, AUTACs utilize small molecules that mimic S-guanylation to tag POIs for K63-linked ubiquitination, triggering their selective degradation via autophagy. AUTOTACs activate the autophagy receptor p62 to facilitate autophagic degradation, while ATTECs use lipidated LC3 to tether polyQ proteins to autophagosomes for their selective removal. LYTACs use glycan tags to mark extracellular proteins for lysosomal degradation. In addition, MoDE-A leverages the lysosomal degradation pathway to eliminate misfolded proteins and protein aggregates. Despite these promising mechanisms, each lysosome-mediated TPD technology faces specific challenges that require further research to optimize its applications. For example, the mechanisms by which AUTACs induce K63 polyubiquitination and ATTECs bind to LC3 remain unclear. AUTOTAC holds promise for activating autophagy cargo receptors for targeted protein degradation, but a more comprehensive understanding is still needed. LYTACs utilize glycan tags to label extracellular POIs, enabling their internalization and degradation within lysosomes through receptor-mediated pathways. In addition, other therapeutic strategies, like AbTACs, PROTABs, GlueTACs, KineTACs, TransTACs, AceTACs, and FRTAC, are emerging to exploit the lysosomal system for degrading disease-associated extracellular and membrane proteins. These approaches utilize diverse targeting elements, including LTRs, transmembrane E3 ligases, IGFIIR, and CPP-LSS sequences. Despite promising advancements, significant challenges remain, including endocytosis efficiency, therapeutic stability, pharmacokinetics, and complex formation. Each TPD method offers unique advantages tailored to specific disease contexts, depending on factors like tissue-specific receptor expression and protein stability. Despite being in the early stages of development, these innovative TPD strategies hold significant potential to revolutionize the treatment of various diseases. Furthermore, academic–industry collaboration is essential to bring these technologies into the biomedicine and expected to be revealed in the near future.

## Figures and Tables

**Figure 1 ijms-26-05582-f001:**
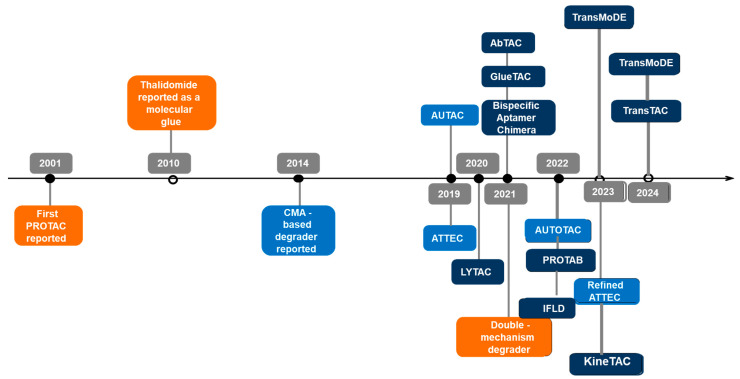
The chronology of development of TPD techniques. Purple represents technologies related to UPS; light blue indicates technologies linked to the autophagy–lysosome pathway; and dark blue represents technologies pertaining to the endosome–lysosome pathway.

**Figure 2 ijms-26-05582-f002:**
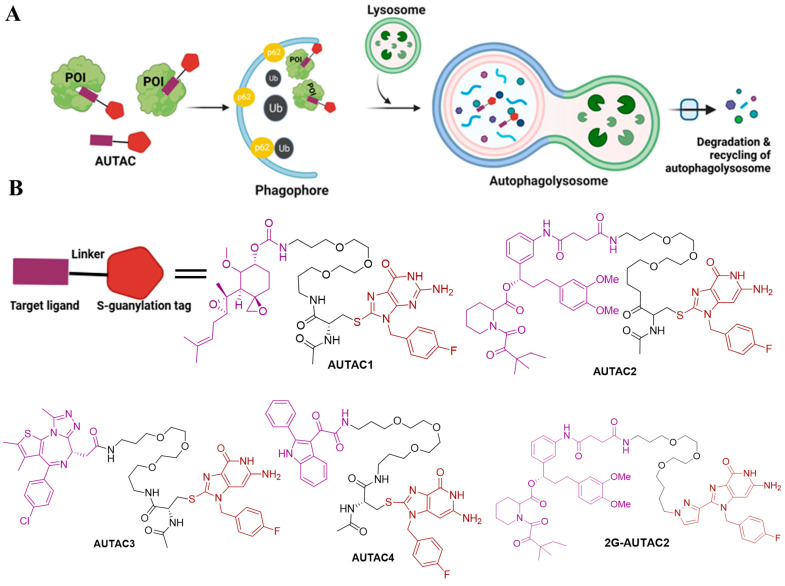
The mechanism of action of ubiquitin-mediated autophagy-targeting degradation: AUTAC. (**A**) A schematic diagram illustrating the mechanism of action of AUTACs. Bifunctional AUTAC compounds bind to the POI and attach a tag that mimics S-guanylation, a post-translational modification that triggers the POI. The POI is subsequently recognized by the autophagy receptor SQSTM1/p62 and directed to the selective autophagy pathway for lysosome-mediated degradation. (**B**) The structures of AUTAC molecules are designed to chemically downregulate various target proteins, including MetAP2, FKBP12, and BET. The modified S- guanylation mimic tags in each AUTACs are highlighted in brown, while the target-specific warheads are marked in violet.

**Figure 3 ijms-26-05582-f003:**
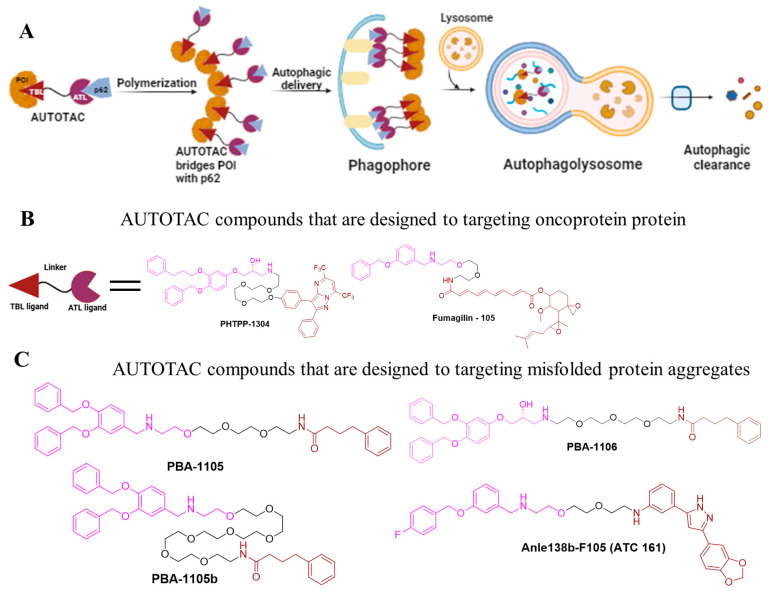
Mechanism of action of macroautophagy-based degrader AUTOTACs. (**A**) Bifunctional AUTOTAC compounds are highly effective in eliminating oncoproteins and aggregated proteins. (**B**) Chemical structure of AUTOTAC molecules designed to degrade oncoproteins, including PHTPP-1304 targeting ERb, vinclozolin M2-2204 targeting AR, and fumagilin-105 targeting. (**C**) AUTOTAC compounds that are designed to target misfolded protein aggregates. PBA-1105, PBA-1106 targeting mutant tau protein, PBA-1105b targeting mutant desmin protein, and Anle 138b-F105 (ATC 161) targeting mutant huntingtin protein.

**Figure 4 ijms-26-05582-f004:**
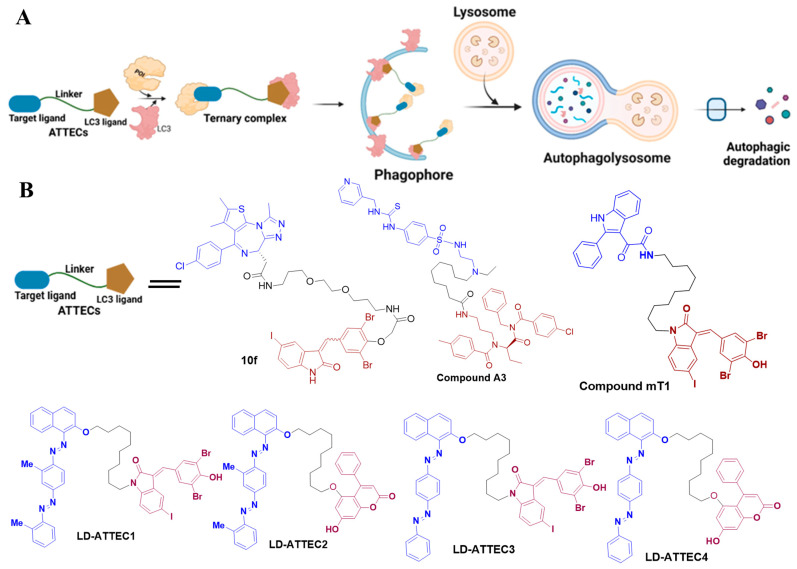
The mechanism of action of targeted autophagy-based degraders, namely ATTECs. (**A**) Bifunctional ATTECs interact with both the POI and LC3, tethering the POI to autophagosomes for subsequent targeted autophagic degradation. (**B**) The chemical structure of ATTECs, including BRD4-ATTEC degrader 10f, NAMPT-ATTEC degrader A3, TSPO-ATTEC degrader mT1, and LD-ATTECs (C1~C4).

**Figure 5 ijms-26-05582-f005:**
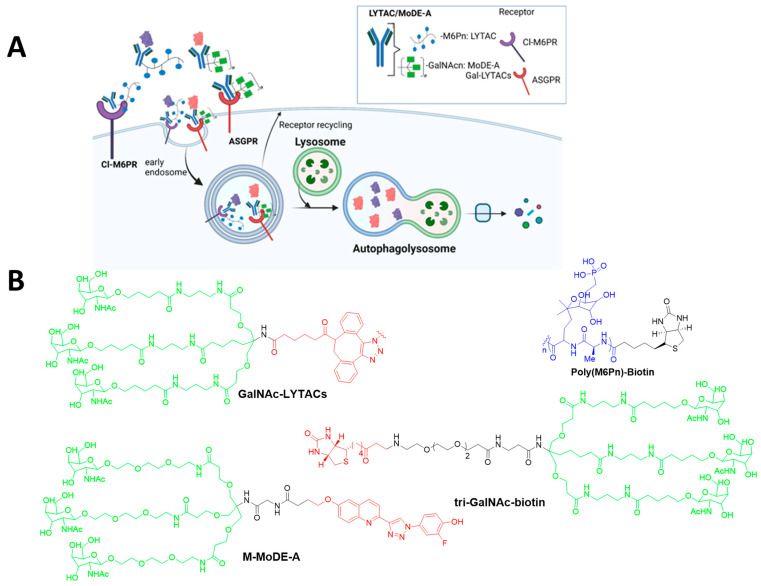
Graphical illustrations of lysosomal-based degradation technologies: LYTACs and MoDE-As. (**A**) LYTACs employ a glycan tag to label an extracellular POI for lysosomal degradation via receptor-mediated internalization. (**B**) The chemical structures of designed LYTACs and MoDE-As-based target-specific ligands.

**Table 1 ijms-26-05582-t001:** Examples of degraders for target extracellular and membrane POI for degradation.

TPD Technology	Compounds	Binding Protein(s)	Target	Activity (DC_50_, or Else Mentioned)	References
AUTACs	AUTAC1	p62 LC3	MetAP2	1 µM in Hela cells	[16]
	AUTAC2	p62 LC3	FKB12	10 µM in Hela cells	[16]
	AUTAC3	p62 LC3	BRD4	-	[16]
	AUTAC4	p62 LC3	TSPO	-	[16]
	2G-AUTAC2	p62 LC3	FKB12	10 µM in Hela cells	[37]
AUTOTACs	PHTPP-1304	p62 LC3	ER	~2 nM	[17]
	VinclozolinM2-2204	p62 LC3	AR	~200 nM	[17]
	Fumagilin-105	p62 LC3	MetAP2	0.7 µM	[17]
	PBA-1105	p62 LC3	tauP301L	1–10 nM	[17]
	PBA-1106	p62 LC3	tauP301L	1–10 nM	[17]
	Anle138b-F105 (ATC 161)	p62 LC3	tauP301L	~3.0 nM	[17]
ATTECs	LD-ATTEC1	LC3	LC3B	*K_d_*~4.2 µM	[19]
	LD-ATTEC2	LC3	LC3B	*K_d_*~4.9 µM	[19]
	LD-ATTEC3	LC3	LC3B	*K_d_*~1.9 µM	[19]
	LD-ATTEC4	LC3	LC3B	*K_d_*~1.3 µM	[19]
	Compound 10f	LC3	BRD4	0.5 µM in MDA-MD-231 and MDA-MB-468 cells	[38]
	Compound A3	LC3	NMPT	-	[39]
	mT1	LC3	TSPO	-	[40]
LYTACs	Poly(M6Pn)-biotein	CI-M6PR	NA-647	-	[20,21]
	GalNAc-LYTACs antibody (Ctx-GN)	ASGPR	EGFR	40% downregulation	[21]
	Tri-GaINAc-biotein	ASGPR	NA-650	-	[41]
MoDE-A	M-MoDE-A	ASGPR	MIF	-	[22]
	D-MoDE-A		a-DNP anti-bodies	NA	[22]
TransMoDEs	TransMoDEs	ASGPR	LRP1	NA	[23]
IFLD	BMS-L1-RGD	*α*v*β*3 integrin	Neutravidin PD-L1 degrader	NA	[24]
AbTAC	R0/Atz	RNF43 or ZNRF3	PD-L1	0.8 nM in HCC2935	[26]
	R3/Atz	RNF43 or ZNRF3	PD-L1	1.2 nM	[26]
	R0/Ctx	RNF43 or ZNRF3	EGFR	5.1 nM	[26]
	R3/Ctx	RNF43 or ZNRF3	EGFR	6.6 nM	[26]
	Z18/Atz	RNF43 or ZNRF3	PD-L1	3.4 nM	[26]
	Z18/Ctx	RNF43 or ZNRF3	EGFR	0.6 nM	[26]
	AC-1	PD-L1	PD-L1	3.4 nM in MDA-MB-231	[26]
PROTAB	PROTAB	RNF43 or ZNRF3	IGF1R/HER2/PD-L1	NA	[26]
GlueTAC	GlueTAC	CPP-LSS	PD-L1	NA	[27]
KineTAC	CXCL12-Atz	CXCR7	PD-L1	NA	[28]
	CXCL12–Tras	‘’	HER2	NA	[28]
	CXCL12–Ctx	‘’	EGFR	NA	[28]
TransTAC	TransTAC	TfR	EGFR, CD20, PD-L1, and a CD19-targeted CAR	NA	[29]
AceTACs	MS182	p62 LC3	mHTT, TDP-43, and Tau	NA	[30]
FRTAC	FRTAC	Folate receptor (FR)	EGFR PD-L1 CD47	NA	[31]

Abbreviation: NA, no data available.

**Table 2 ijms-26-05582-t002:** Advantages and limitation of emerging degrader technologies.

Degrader Properties	Potential Targets	Type of Proteins Degraded	Degradation System/Pathway	Advantages	Limitations
PROTAC	Tethers to E3 ubiquitin ligase and triggers K48 polyubiquitination	Short-lived and soluble, misfolded	Ubiquitin–proteasomal system (UPS)/proteasome	Selective protein degradation; clear mechanisms of action; relatively high selectivity; catalytic mode of action	E3-ligase- and ubiquitination-dependent; undesirable PK/PD profile; low BBB penetration capability
AUTAC	Triggers K63 polyubiquitin and autophagosome	Long-lived; insoluble protein aggregates	Selective macroautophagy–lysosomal	Potential broad target; independent proteasome; ability to degrade mitochondria	Unclear mechanism of action; dependent on K63 ubiquitination; possible selective autophagy
AUTOTAC	p62/SQSRM1, one of the selective autophagy receptors	Long-lived; insoluble protein aggregates	Selective macroautophagy–lysosomal	Broad target scope leverages autophagy pathway; selective degradation; reduced off-target effects	Dependency on autophagy pathway; limited understanding of mechanisms; off-target effects of AUTOTACs
ATTEC	LC-3, a key protein of autophagosome	Long-lived; insoluble protein aggregates	Macroautophagy–lysosomal	Potentially a broad target spectrum; direct targeting to the degradation machinery; potentially effective in all cell types; low molecular weight	The LC3-bound chemical moieties need to be solved; lack of studies on designed chimeras
LYTAC	Lysosome-targeting receptors (LTRs)	Extracellular; membrane; cell-surface protein	Endosome–lysosomal pathway for degradation of glycosylated proteins	Applicable to extracellular and transmembrane proteins; UPS/proteosomal-independent degradation	Not applicable to cytosolic proteins; large molecular weight and poor permeability; possible immunogenicity as it uses peptide-like structures
GalNAc-LYTAC	LTR (ASGPR)	Extracellular; membrane; cell-surface protein	LTR (ASGPR)-mediated internalization and lysosomal delivery	Selective membrane protein degradation; preserves lysosomal integrity	Modulate PK properties to control off-target clearance
MoDE-As	ASGPR	Extracellular; membrane; cell-surface proteins	Endosome–lysosomal pathway	Selective targeting of extracellular proteins; proteasome independent degradation; relatively small in size	Limited intracellular targets; challenges of specific cellular targets; short half-life in vivo
TransMoDEs	ASGPR	Extracellular; membrane	lysosomal	Hepatocyte-specific degradation; ubiquitin–proteasome independent	Limited to receptor-expressing tissues and extracellular targets; early-stage development
IFLD	αvβ3 integrin	Extracellular and membrane proteins	RGD peptide: integrin-mediated endocytosis	Efficient degradation of extracellular and membrane proteins	Dependence on integrin expression; risk of off-target internalization; early-stage research
AbTAC/PROTAB	E3-ligase	Extracellular and membrane proteins	RNF43- or ZNRF3-mediated lysosomal degradation	Targets membrane proteins; high specificity vs. antibody	High cost; potential immune response in vivo; unclear endocytosis mechanism; Hook effect
GlueTAC	CPP-LSS	Extracellular and membrane proteins	CPP-LSS-mediated internalization and lysosomal degradation	Targets membrane proteins; independent of specific receptors or E3 ligase; irreversible attachment of covalent nanobodies to surface POI	Safety profiles of GlueTACs containing unnatural amino acid half-life needs to be determined
KineTAC	CXCL7	CXCL12: CXCR7-mediated endocytosis	Lysosome	Degrades membrane proteins with intracellular domains and small molecules	Targets few membrane proteins; binds intracellular site of membrane protein and access; UPS in cytosol; importance of linker on binding
TransTAC	TfR protein	Transmembrane proteins	Lysosome	Proteasome-independent Target degradation of transmembrane proteins	Poor drug-like properties; Early-stage research; delivery and design complexity
IFLD	αvβ3 integrin	Extracellular and membrane proteins	RGD peptide: integrin-mediated endocytosis	Efficient degradation of extracellular and membrane proteins	Dependence on integrin expression; risk of off-target internalization; early-stage research
AceTACs	LC3 on the autophagosome membrane	Extracellular and membrane proteins	Lysosome	Proteasome-independent; selective degradation of the target-specific substrates; lower off-target degradation	Poor oral bioavailability and permeability; early-stage research
FRTAC	Folate receptor	Extracellular and membrane proteins	Lysosome	Cancer cell selectivity improved precision and efficiency	Off-target effects; unclear mechanism; early-stage research; hook effects

## Data Availability

No new data were created or analyzed in this study.

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
