# Peer review of "Emerging Concepts of Targeted Protein Degrader Technologies via Lysosomal Pathways"

_ijms, 2025, doi:10.3390/ijms26125582_

Round 1
Reviewer 1 Report
Comments and Suggestions for Authors
This review discusses recent progress in lysosome-targeting degradation technologies. The authors provide an overview of the history of targeted protein degradation (TPD), briefly comparing PROTACs with recent lysosomal-targeting degraders in terms of mechanisms and targets. They present a detailed discussion on autophagy-recruiting degraders, followed by a brief mention of degraders for extracellular or membrane proteins. However, the paper does not cover the most up-to-date research in this field and contains misunderstandings regarding lysosomal degradation pathways. The manuscript should be considered for acceptance after addressing the following major issues:
- The main issue is that it is not clear what this paper is trying to cover. The title, "Emerging New Concepts of Targeted Protein Degrader Technologies via Lysosomal Pathways," suggests that the paper should provide a comprehensive discussion of all recently discovered lysosomal-targeting degraders, including both endosome- and autophagy-recruiting degraders. However, the authors primarily focus on small molecule-based autophagy-involving degraders. They do not discuss some of the recent studies, such as autophagy receptor-inspired targeting chimeras (AceTACs). Moreover, they only cover some of the pioneering work on lysosomal-targeting degraders that utilize the endosome/lysosome pathway. Many key recent discoveries in this field, such as KineTAC, TransTAC, and FRTAC, are not included in this paper. It is hard to identify the scope of this paper.
- There is a misunderstanding about lysosomal degradation pathways. Not all lysosomal degradation pathways involve autophagy. Autophagy is a major pathway that delivers cytoplasmic material to the lysosome for degradation. However, extracellular materials or membrane proteins can be delivered to the lysosome through the endosome without fusing with autophagosomes. Most lysosome-targeting degraders for extracellular or membrane proteins utilize the endosome/lysosome pathway rather than the autophagy/lysosome pathway. Specifically, Page 1, line 41: It would be more appropriate to state that the UPS and lysosomal degradation pathways are the two major pathways for TPD and then introduce the biology of the endosome/lysosome pathway. Page 4, line 133: LYTAC and MoDE-As do not go through the autophagy pathway. Figure 5a: The illustration should not include autophagy.
- Page 11, line 348: The first LYTAC paper not only used NA as the target but also explored other membrane proteins. The authors need to provide a more detailed discussion of this paper.
- Page 13, line 363: The term “the study” is unclear. Is it referring to the same study mentioned earlier or a different one? Please specify.
- It is not appropriate to classify MoDE-As as a separate category, as they follow the same mechanism as LYTAC. Additionally, if the authors intend to focus solely on small molecule degraders, they should include more recent studies, such as TransMoDEs (Transcytosis-inducing molecular degraders of extracellular proteins) and the integrin-facilitated lysosomal degradation (IFLD) strategy.
- Figures 2-5: The resolution is low.
- Page 3, line 91: The full name of CMA should be provided.
- Page 13, line 360: There is a typo—"ASGPR" should be corrected.
Comments on the Quality of English Language
The English could be improved to more clearly express the research.
Author Response
Response to Reviewers #1’ comments
Reviewer #1: This review discusses recent progress in lysosome-targeting degradation technologies. The authors provide an overview of the history of targeted protein degradation (TPD), briefly comparing PROTACs with recent lysosomal-targeting degraders in terms of mechanisms and targets. They present a detailed discussion on autophagy-recruiting degraders, followed by a brief mention of degraders for extracellular or membrane proteins. However, the paper does not cover the most up-to-date research in this field and contains misunderstandings regarding lysosomal degradation pathways. The manuscript should be considered for acceptance after addressing the following major issues:
We would like to thank the reviewer for their thoughtful comments and concern, particularly regarding the observation that the manuscript does not reflect the most up-to-date research in the field and includes misunderstandings related to lysosomal degradation pathways. Our original intention was to focus preliminary on the autophagy–lysosome pathway. However, in response to the reviewer’s feedback, we have expanded the scope of the manuscript to incorporate the latest research on the both the autophagy–lysosome and endosome–lysosome pathways.
Q1. The main issue is that it is not clear what this paper is trying to cover. The title, "Emerging New Concepts of Targeted Protein Degrader Technologies via Lysosomal Pathways," suggests that the paper should provide a comprehensive discussion of all recently discovered lysosomal-targeting degraders, including both endosome- and autophagy-recruiting degraders. However, the authors primarily focus on small molecule-based autophagy-involving degraders. They do not discuss some of the recent studies, such as autophagy receptor-inspired targeting chimeras (AceTACs). Moreover, they only cover some of the pioneering work on lysosomal-targeting degraders that utilize the endosome/lysosome pathway. Many key recent discoveries in this field, such as KineTAC, TransTAC, and FRTAC, are not included in this paper. It is hard to identify the scope of this paper.
We appreciate the reviewer's valuable feedback. We have revised the review title “Emerging Concepts of Targeted Protein Degrader Technologies via lysosomal pathways” and have included an extensive discussion of recently identified key lysosomal-targeting degraders, incorporating both endosome- and autophagy-recruiting degraders.
Q2. There is a misunderstanding about lysosomal degradation pathways. Not all lysosomal degradation pathways involve autophagy. Autophagy is a major pathway that delivers cytoplasmic material to the lysosome for degradation. However, extracellular materials or membrane proteins can be delivered to the lysosome through the endosome without fusing with autophagosomes. Most lysosome-targeting degraders for extracellular or membrane proteins utilize the endosome/lysosome pathway rather than the autophagy/lysosome pathway. Specifically, Page 1, line 41: It would be more appropriate to state that the UPS and lysosomal degradation pathways are the two major pathways for TPD and then introduce the biology of the endosome/lysosome pathway. Page 4, line 133: LYTAC and MoDE-As do not go through the autophagy pathway. Figure 5a: The illustration should not include autophagy.
We appreciate the reviewer's valuable feedback. We have revised the sentences based on reviewer’s feedback in Page 1, line 41 sentences: Accordingly, we have explain the UPS and lysosomal degradation pathways are the two major pathways for TPD and further introduce the biology of the endosome/lysosome pathway.
Q3. Page 11, line 348: The first LYTAC paper not only used NA as the target but also explored other membrane proteins. The authors need to provide a more detailed discussion of this paper.
We have revised the section on the LYTAC protein mechanism and provided a more detailed explanation.
Q4. Page 13, line 363: The term “the study ” is unclear. Is it referring to the same study mentioned earlier or a different one? Please specify.
We have corrected the sentence.
Q5. It is not appropriate to classify MoDE-As as a separate category, as they follow the same mechanism as LYTAC. Additionally, if the authors intend to focus solely on small molecule degraders, they should include more recent studies, such as TransMoDEs (Transcytosis-inducing molecular degraders of extracellular proteins) and the integrin-facilitated lysosomal degradation (IFLD) strategy.
We appreciate the reviewer's concerns; we have included the more recent studies such as TransMoDEs and IFLD strategy as reviewer suggested.
Q6, Figures 2-5: The resolution is low.
We have enhanced the resolution of figure 2-5.
Q7. Page 3, line 91: The full name of CMA should be provided.
We have updated the full name of CMA
Q8. Page 13, line 360: There is a typo—"ASGPR" should be corrected.
We have corrected to Typo error
Reviewer 2 Report
Comments and Suggestions for Authors
- p2 line 73, it is recommended that the authors include the full name when abbreviations first show up (MoDE-A).
- Section 3. New Frontiers in Small Molecule TPD. Could authors consider rephrasing the section title as this can be misleading: the section talks about the autophagy pathway but the title seems to focus on small molecules entities
- Can authors possibly improve the quality of the image? In case of the image was adopted from other sources, a statement of adoption is recommended.
- In this manuscript, Alam et al. gave a thorough review on the current progress of targeted protein degradation (field), focusing on the advancement made by novel degraders leveraging the autophagy mechanisms. Alam et al. covered the mechanism, history, and listed reported molecules including ATTAC, AUTOTAC, LYTAC, AbTAC, MoDE-A etc., summarizing this emerging field from origination to modern development in a concise way. Given the recently increased interests in novel TPD methods and the way Alam et al. presented the data, the manuscript is recommended for publication with minor corrections on details.
- The conventional TPD heavily relies on the ubiquitin system (UPS) that still suffers from resistance and restricted the target within the intracellular proteins. Therefore, there is more interest to understand how such issues can be resolved or circumvented. This review presented autophagy-targeting degraders, which is a UPS-independent cellular machinery and could overcome the challenges presented by the conventional TPDs. Therefore, this review can be considered as filling the gap in the field as it framed out the whole research field in a concise and accurate manner.
- The conclusions Alam et al. proposed is in line with the current advancement of TPD field regardless of various mechanisms. The conclusion that autophagy-targeting degraders could stand out as novel and more effective therapeutics is well supported and rationalized by the data incorporated in earlier sections.
- All references are appropriate, and no methodology improvement is necessary for the manuscript to be published.
Author Response
Response to Reviewers#2’ comments
We would like to thank the Reviewer suggestion for improvement of manuscript. We have extensively revised the manuscript based Reviewer comments and addressed all point which Reviewer has mention in below.
Q1: p2 line 73, it is recommended that the authors include the full name when abbreviations first show up (MoDE-A).
We have updated the full name of MoDE-A.
Q2. Section 3. New Frontiers in Small Molecule TPD. Could authors consider rephrasing the section title as this can be misleading: the section talks about the autophagy pathway but the title seems to focus on small molecules entities.
In response to reviewer’s comment, we have updated the New Frontiers in Small Molecule TPD to New Frontiers in Lysosome-mediated TPD.
Q3. Can authors possibly improve the quality of the image? In case of the image was adopted from other sources, a statement of adoption is recommended.
In response to reviewer’s comment, we have improved the image quality in the manuscript and mentioned the statement of adoption in acknowledgment section.
Q4. In this manuscript, Alam et al. gave a thorough review on the current progress of targeted protein degradation (field), focusing on the advancement made by novel degraders leveraging the autophagy mechanisms. Alam et al. covered the mechanism, history, and listed reported molecules including ATTAC, AUTOTAC, LYTAC, AbTAC, MoDE-A etc., summarizing this emerging field from origination to modern development in a concise way. Given the recently increased interests in novel TPD methods and the way Alam et al. presented the data, the manuscript is recommended for publication with minor corrections on details. The conventional TPD heavily relies on the ubiquitin system (UPS) that still suffers from resistance and restricted the target within the intracellular proteins. Therefore, there is more interest to understand how such issues can be resolved or circumvented. This review presented autophagy-targeting degraders, which is a UPS-independent cellular machinery and could overcome the challenges presented by the conventional TPDs. Therefore, this review can be considered as filling the gap in the field as it framed out the whole research field in a concise and accurate manner. The conclusions Alam et al. proposed is in line with the current advancement of TPD field regardless of various mechanisms. The conclusion that autophagy-targeting degraders could stand out as novel and more effective therapeutics is well supported and rationalized by the data incorporated in earlier sections.
All references are appropriate, and no methodology improvement is necessary for the manuscript to be published.
We would like to express our great thanks and appreciation to the reviewer for his review, opinion and praise of our work. In response to the reviewer’s comment, we have updated the manuscript.
Round 2
Reviewer 1 Report
Comments and Suggestions for Authors
The authors have addressed all the concerns.